# ElimiNet: A Model for Eliminating Options for Reading Comprehension with Multiple Choice Questions

## Abstract

The task of Reading Comprehension with Multiple Choice Questions, requires a human (or machine) to read a given {*passage, question*} pair and select one of the $n$ given options. The current state of the art model for this task first computes a query-aware representation for the passage and then *selects* the option which has the maximum similarity with this representation. However, when humans perform this task they do not just focus on option selection but use a combination of *elimination* and *selection*. Specifically, a human would first try to eliminate the most irrelevant option and then read the document again in the light of this new information (and perhaps ignore portions corresponding to the eliminated option). This process could be repeated multiple times till the reader is finally ready to select the correct option. We propose *ElimiNet*, a neural network based model which tries to mimic this process. Specifically, it has gates which decide whether an option can be eliminated given the {*document, question*} pair and if so it tries to make the document representation orthogonal to this eliminatedd option (akin to ignoring portions of the document corresponding to the eliminated option). The model makes multiple rounds of partial elimination to refine the document representation and finally uses a selection module to pick the best option. We evaluate our model on the recently released large scale RACE dataset and show that it outperforms the current state of the art model on 7 out of the 13 question types in this dataset. Further we show that taking an ensemble of our *elimination-selection* based method with a *selection* based method gives us an improvement of 7% (relative) over the best reported performance on this dataset.

## 1 Introduction

Reading comprehension is the task of answering questions from a given passage. An AI agent which can display such capabilities would be useful in a wide variety of commercial applications such as answering questions from financial reports of a company, troubleshooting using product manuals, answering general knowledge questions from Wikipedia documents, *etc*. Given its widespread applicability, several variants of this task have been studied in the literature. For example, given a passage and a question, the answer could either (i) match some span in the passage or (ii) be synthesized from the passage or (iii) be one of the $n$ given candidate answers. The last variant is typically used in various high school, middle school and competitive examinations. We refer to this as Reading Comprehension with Multiple Choice Questions (RC-MCQ). There is an increasing interest in building AI agents with deep language understanding capabilities which can perform at par with humans on such competitive tests. For example, recently Lai et al. (2017) have released a large scale dataset for RC-MCQ collected from Chinese high school and middle school English examinations comprising of 28000 passages and 100000 questions. The large size of this dataset makes it possible to train and evaluate complex neural network based models and measure the scientific progress on RC-MCQ.

While answering such Multiple Choice Questions (MCQs), humans typically use a combination of *option elimination* and *option selection*. More specifically, it makes sense to first try to eliminate options which are completely irrelevant for the given question. While doing so, we may also be able to discard certain portions of the document which are not relevant to the question (because they

revolve around the option which has been eliminated). This process can then be repeated multiple times, each time eliminating an option and refining the document (by discarding irrelevant portions). Finally, when it is no longer possible to eliminate any option, we can pick the best option from the remaining options. In contrast, current state of the art models for RC-MCQ focus explicitly on option selection. Specifically, given a question and a passage, they first compute a question aware representation of the passage (say $d_q$). They then compute a representation for each of the $n$ options and select an option whose representation is closest to $d_q$. There is no iterative process where options get eliminated and the representation of the document gets refined in the light of this elimination.

We propose a model which tries to mimic the human process of answering MCQs. Similar to the existing state of the art method (Dhingra et al., 2017), we first compute a query-aware representation of the document (which essentially tries to retain only those portions of the document which are relevant to the question). We then use an elimination gate which takes a soft decision as to whether an option needs to be eliminated or not. This gate depends on the question, document and option. Next, akin to the human process described above, we would like to discard portions of the document representation which are aligned with this eliminated option. We do this by subtracting the component of the document representation along the option representation (same as Gram-Schmidt orthogonalization). The amount of orthogonalization depends on the soft decision given by the elimination gate. We repeat this process multiple times, during each pass doing a soft elimination of the options and refining the document representation. At the end of a few passes, we expect the document representation to be orthogonal (hence dissimilar) to the irrelevant options. Finally, we use a selection module to select the option which is most similar to the refined document representation. We refer to this model as *ElimiNet*.

We evaluate *ElimiNet* on the RACE dataset and compare it with Gated Attention Reader (GAR) (Dhingra et al., 2017) which is the current state of the art method on this dataset. We show that of the 13 question types in this dataset our model outperforms GAR on 7 question types. We also visualize the soft elimination probabilities learnt by *ElimiNet* and observe that it indeed learns to iteratively refine the document representation and push the probability mass towards the correct option. Finally, we show that an ensemble model combining *ElimiNet* with *GAR* gives an accuracy of 47.2% which is 7% (relative) better than the best reported performance on this dataset.

## 2 RELATED WORK

Over the fast few years, the availability of large scale datasets has led to renewed interest in the task of Reading Comprehension. These datasets cover different variations of the Reading comprehension task. For example, the SQuAD (Rajpurkar et al., 2016), TriviaQA (Joshi et al., 2017), NewsQA (Trischler et al., 2016), etc contain {*question, answer, passage*} where the answer matches a span of the passage. On the other hand, CNN/DailMaily (Hermann et al., 2015), Children's Book Test(CBT) (Hill et al., 2015) and Who Did What(WDW) dataset (Onishi et al., 2016) offer cloze-style RC where the task is to predict a missing word/entity in the question. Some other datasets such as MCTest (Richardson et al., 2013), AI2 (Khashabi et al., 2016) and RACE contain RC with multiple choice questions (RC-MCQ) where the task is to select the right answer.

The advent of these datasets and the general success of deep learning for various NLP tasks, has led to a proliferation of neural network based models for RC. For example, the models proposed in (Xiong et al., 2016; Seo et al., 2016; Wang et al., 2017; Hu et al., 2017) address the first variant of RC requiring span prediction as in the SQuAD dataset. Similarly, the models proposed in (Chen et al., 2016; Kadlec et al., 2016; Cui et al., 2017; Dhingra et al., 2017) address the second variant of RC requiring cloze-style QA. Finally, Lai et al. (2017) adapt the the models proposed in (Chen et al., 2016; Dhingra et al., 2017) for cloze-style RC and use them to address the problem of RC-MCQ. Irrespective of which of the three variants of RC they address, these models use a very similar framework. Specifically, these models contain components for (i) encoding the passage (ii) encoding the query (iii) capturing interactions between the query and the passage (iv) capturing interactions between query and the options (for MCQ) (v) making multiple passes over the passage and (vi) a decoder to predict/generate/select an answer. The differences between the models arise from the specific choice of encoder, decoder, interaction function and iteration mechanism. Most of the current state of the art models can be seen as special instantiations of the above framework.

The key difference between our model and existing models for RC-MCQ is that we introduce components for (soft-)eliminating irrelevant options and refining the passage representation in the light of this elimination. The passage representation thus refined over multiple (soft-)elimination rounds is then used for selecting the most relevant option. To the best of our knowledge, this is the first model which introduces the idea of option elimination for RC-MCQ.

## 3 PROPOSED MODEL

Given a passage $D = [w_1^d, w_2^d, \ldots, w_M^d]$ of length $M$, a query $Q = [w_1^q, w_2^q, \ldots, w_N^q]$ of length $N$ and $n$ options $Z_k = [w_1^z, w_2^z, \ldots, w_{J_k}^z]$ where $1 \leqslant k \leqslant n$ and each option is of length $J_k$, the task is to predict a conditional probability distribution over the options (*i.e.*, to predict $P(Z_i|D,Q)$). We model this distribution using a neural network which contains modules for encoding the question/passage/options, capturing the interactions between them, eliminating options and finally selecting the correct option. We refer to these as the encoder, interaction, elimination and selection modules. Among these, the main contribution of our work is the introduction of a module for elimination. Specifically, we introduce a module to (i) decide whether an option can be eliminated (ii) refine the passage representation to account for eliminated/uneliminated options and (iii) repeat this process multiple times. In the remainder of this section we describe the various components of our model.

**Encoder Module:** We first compute vectorial representations of the query and options. We do so by using a bidirectional recurrent neural network which contains two Gated Recurrent Units (GRU) (Chung et al., 2014), one which reads the given string (query or option) from left to right and the other which reads the string from right to left. For example, given the query $Q = [w_1^q, w_2^q, \ldots, w_N^q]$, each GRU unit computes a hidden representation for each time-step (word) as:

$$\overrightarrow{h_i^q} = \overrightarrow{GRU_q}(\overrightarrow{h_{i-1}^q}, e(w_i^q))$$
$$\overleftarrow{h_i^q} = \overleftarrow{GRU_q}(\overleftarrow{h_{i-1}^q}, e(w_i^q))$$

where $e(w_i^q) \in \mathbb{R}^d$ is the d-dimensional embedding of the query word $w_i^q$. The final representation of each query word is a concatenation of the forward and backward representations (*i.e.*, $h_i^q = [\overleftarrow{h_i^q}, \overrightarrow{h_i^q}]$). Similarly, we compute the bi-directional representations for each word in each of the $k$ options as $h_i^{z_k} = [\overleftarrow{h_i^{z_k}}, \overrightarrow{h_i^{z_k}}]$. Just to be clear, $h_i^{z_k}$ is the representation of the $i$-th word in the $k$-th option ($z_k$). We use different GRU cells for the query and options. The same GRU cell is used for computing the representation of all the $n$ options. Note that the encoder also computes a representation of each passage word as simply the word embedding of the passage word (*i.e.*, $h_i^d = e(w_i^d)$). Later on in the interaction module we use a GRU cell to compute the interactions between the passage words.

**Interaction Module:** Once the basic query and passage word representations have been computed, the idea is to allow them to interact so that the passage words' representations can be refined in the light of the query words' representations. This is similar to how humans first independently read the passage and the question and then read the passage multiple times trying to focus on the portions which are relevant to the question. To achieve this, we use the same multi-hop architecture for iteratively refining passage representations as proposed in Gated Attention Reader (Dhingra et al., 2017). At each hop $t$, we use the following set of equations to compute this refinement:

$$\alpha_i^t = \text{softmax}(Q^T d_i^t)$$

where, $Q \in \mathbb{R}^{N \times l}$ is a matrix whose columns are $h_1^q, h_2^q, ..., h_N^q$ as computed by the encoder. $\alpha_i^t \in \mathbb{R}^N$ such that each element $j$ of $\alpha_i^t$ essentially computes the importance of the $j$-th query word for the $i$-th document word during hop $t$. At the 0-th hop, $d_i^0 = h_i^d = e(w_i^d) \in \mathbb{R}^l$ is simply the embedding of the $i$-th passage word. The goal is to refine this embedding over each hop based on interactions with the query. Next, we compute,

$$\tilde{q}_i^t = Q\alpha_i^t$$

where $\tilde{q}_i^t \in \mathbb{R}^l$ computes the importance of each dimension of the current passage word representation and is then used as a gate to scale up or scale down different dimensions of the passage word

representation.

$$\tilde{d}_i^t = d_i^t \odot \tilde{q}_i^t$$

We now allow these refined passage word representations to interact with each other using a bi-directional recurrent neural network to compute $d_i^{(t+1)}$ for the next hop.

$$\overrightarrow{d_i^{(t+1)}} = \overrightarrow{\text{GRU}}_D^{(t+1)}(\overrightarrow{d_{i-1}^{(t+1)}}, \tilde{d}_i^{(t)})$$
$$\overleftarrow{d_i^{(t+1)}} = \overleftarrow{\text{GRU}}_D^{(t+1)}(\overleftarrow{d_{i-1}^{(t+1)}}, \tilde{d}_i^{(t)})$$
$$d_i^{(t+1)} = [\overleftarrow{d_i^{(t+1)}}, \overrightarrow{d_i^{(t+1)}}]$$

The above process is repeated for $T$ hops wherein each hop takes $d_i^{(t)}, Q$ as the input and computes a refined representation $\tilde{d}_i^{(t+1)}$. After $T$ hops, we obtain a fixed-length vector representation of the passage by combining the passage word representations using a weighted sum.

$$m_i = \text{softmax}(\tilde{d}_i^{(T)} W_{att} h_N^q)$$
$$x = \sum_{i=1}^{M} m_i \tilde{d}_i^{(T)} \tag{1}$$

where $m_i$ computes the importance of each document word and $x$ is a weighted sum of the document representations.

**Elimination Module:**  The aim of the elimination module is to refine the passage representation so that it does not focus on portions which correspond to irrelevant options. To do so we first need to decide whether an option can be eliminated or not and then ensure that the passage representation gets modified accordingly. For the first part we introduce an *elimination* gate to enable a soft-elimination.

$$e_i = \text{sigmoid}(W_e x + V_e h^q + U_e h^{z_i})$$

Note that this gate is computed separately for each option $i$. In particular, it depends on the final state of the bidirectional option GRU ($h^{z_i} = h_{J_i}^{z_i}$). It also depends on the final state of the bidirectional query GRU ($h^q = h_N^q$) and the refined document representation ($x$) computed by the interaction module. $W_e, V_e, U_e$ are parameters which will be learned.

Based on the above soft-elimination, we want to now refine the document representation. For this, we compute $x_i^e$ which is the component of the document representation ($x$) orthogonal to the option representation ($h^{z_i}$) and $x_i^r$ which is the component of the passage representation along the option representation.

$$r_i = \frac{<x, h^{z_i}> h^{z_i}}{|x|^2}$$
$$x_i^e = x - r_i \tag{2}$$
$$x_i^r = x - x_i^e \tag{3}$$

The *elimination gate* then decides how much of $x_i^e$ and $x_i^r$ need to be retained.

$$\tilde{x}_i = e_i \odot x_i^e + (1 - e_i) \odot x_i^r$$

If $e_i = 1$ (eliminate) then the document representation will be made orthogonal to the option representation (akin to ignoring portions of the document relevant to the option) and $e_i = 0$ (don't eliminate) then the document representation will be aligned with the option representation (akin to focusing on portions of the document relevant to the option).

Note that in equations (2) and (3) we completely subtract the components along or orthogonal to the option representation. We wanted to give the model some flexibility to decide how much of this component to subtract. To do this we introduce another gate, called the subtract gate,

$$s_i = \text{sigmoid}(W_s x + V_s h^q + U_s h^{z_i})$$

where $W_s, V_s, U_s$ are parameters that need to be learned. We then replace the RHS of Equations 2 and 3 by $x - s_i \odot r_i$ and $x - s_i \odot x_i^e$. Thus the components $r_i$ and $r_i^\perp$ used in Equation (2) and (3) are gated using $s_i$. One could argue that $c_i$ itself could encode this information but in practice we found that separating these two functionalities (elimination and subtraction) works better.

For each of the $n$ options, we independently compute representations $\tilde{x}_1, \tilde{x}_2, ...\tilde{x}_n$. We then combine these $n$ representations to obtain a single refined representation for the document.

$$b_i = v_b^T \tanh(W_b \tilde{x}_i + U_b h^{z_i})$$
$$\beta_i = \text{softmax}(b_i)$$
$$\tilde{x} = \sum_{i=1}^{n} \beta_i \tilde{x}_i \tag{4}$$

Note that $\tilde{x}_1, \tilde{x}_2, ...\tilde{x}_n$ represent the $n$ option-specific document representations and $\beta_i$'s give us a way of combining these option specific representations into a single document representation. We repeat the above process for $L$ hops wherein the $m$-th hop takes $\tilde{x}^{m-1}$, $h_q$ and $h_{z_i}$ as input and returns a refined $\tilde{x}^m$ computed using the above set of equations.

**Selection Module**    Finally, the selection module takes the refined document representation $\tilde{x}^L$ after $L$ elimination hops and compute its bilinear similarity with each option representation.

$$\text{score}(i) = \tilde{x}^L W_{att} h^{z_i}$$

where $\tilde{x}^L$ and $h^{z_i}$ are vectors and $W_{att}$ is a matrix which needs to be learned. We select the option which gives the highest score as computed above. We train the model using the cross entropy loss by normalizing the above scores (using softmax) first to obtain a probability distribution.

## 4    EXPERIMENTAL SETUP

In this section we describe the dataset used for evaluation, the hyperparameters of our model, training procedure and other state of the art models used for comparison.

**Dataset:**    We evaluate our model on the recently released RACE dataset which contains multiple choice questions collected from Chinese high school and middle school English examinations comprising of 28000 passages and 100000 questions. The high school portion of the dataset referred to as RACE-H contains 62445, 3451 and 3498 questions for training, validation and testing respectively. Similarly, the middle school portion of the dataset referred to as RACE-M contains 18728, 1021 and 1045 questions for training, validation and testing respectively.
This dataset contains a wide variety of questions of varying degrees of complexity. For example, some questions ask for the most appropriate title for the passage which requires deep language understanding capabilities to comprehend the entire passage. There are some questions which ask for the meaning of a specific term or phrase in the context of the passage. Similarly, there are some questions which ask for the key idea in the passage. Finally, there are some standard Wh-type questions. Given this wide variety of questions, we wanted to see if there are specific types of questions for which an elimination module makes more sense. To do so, we categorize the questions in the test dataset into 13 categories. The distribution of questions belonging to each of these categories in RACE-H and RACE-M are shown in Figure 1.

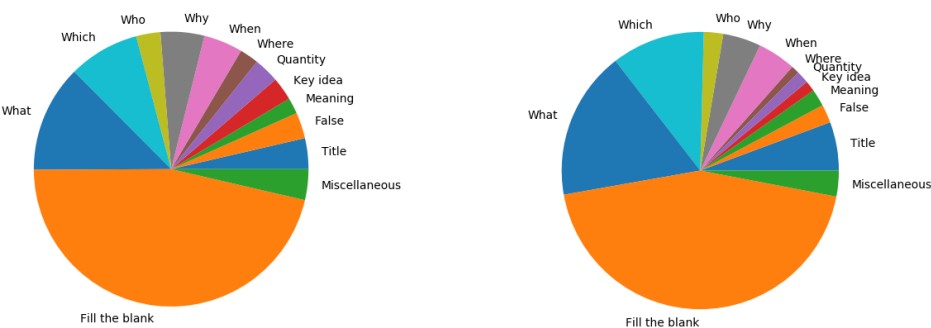

Figure 1: Distribution of different question types in the RACE-Mid (left) and RACE-High (right) portions of the dataset

**Training Procedures:** We tried two different ways of training the model. In the first case, we train the parameters of all the modules (encoder, interaction, elimination and selection) together. In the second case, we first remove the elimination module and train the parameters of the remaining modules. We then fix the parameters of the encoder and interaction module and train only the elimination and selection module. The idea was to give a better chance to the elimination module to learn to refine the document representations (in other words, ensure that the entire learning is focused on the elimination module). Of course, we also had to learn the parameters of the selection module from scratch because it now needs to work with the refined document representations. In practice, we found that this pre-training step did not improve the performance by much. Hence, we report results only for the first case (*i.e.*, end-to-end training).

**Hyperparameters:** We restrict our vocabulary to the top 50K words appearing in the passage, query and options in the dataset. We use the same vocabulary for the passage, query and options. We do not have any restrictions on the passage length as most of the questions require the reader/machine to understand the whole passage before answering the question (for example, "What is the title of the passage?"). In other words, we do not trim the passage but pass the entire passage through our bi-directional GRU.

We use the same train, valid, test splits as provided by the authors. We tune all our model based on the accuracy achieved on the validation set. We initialize the word embeddings with 100 dimensional Glove embeddings. We experimented with both fine-tuning and not fine-tuning these word embeddings. The hidden size for BiGRU is the same across the passage, query and document and we considered the following sizes :$[64, 128, 256]$. We experiment with $[1, 2, 3]$ hops in the interaction module and $[1, 3, 6]$ passes in the elimination module. We add dropout at the input layer to the BiGRUs and experimented with dropout values of $[0.2, 0.3, 0.5]$. We tried both Adam and SGD as the optimizer. For Adam, we set the learning rate to $10^{-3}$ and for SGD we tried learning rates of $[0.1, 0.3, 0.5]$. In general, we found that Adam converges much faster. We train all our models for a maximum of 50 epochs as we did not see any benefit of training beyond 50 epochs.

**Models compared:** We compare our results with the current state of the art model on RACE dataset, namely, Gated Attention Reader (Dhingra et al., 2017). Note that this models was initially proposed for cloze-style RC and is in fact the current state of the art model for cloze-style RC. The authors of RACE dataset adapted this model for RC-MCQ by replacing the output layer with a layer which computes the bilinear similarity between the option representation and document representation.

## 5 RESULTS AND DISCUSSIONS

In this section, we discuss the results of our experiments as described above.

## 5.1 PERFORMANCE OF INDIVIDUAL MODELS

We compare the accuracy of different models on RACE-Mid (middle school), RACE-High (high school) and full RACE testset comprising of both RACE-Mid and RACE-High. For each dataset, we compare the accuracy for each question type. These results are summarized in Figure 2. We observe that, on RACE-Mid ElimiNet performs better than Gated Attention Reader (GAR) on 9 out of 13 categories. Similarly, on RACE-High ElimiNet performs better than GAR on 6 out of 13 categories. Finally, on RACE-full, ElimiNet performs better than GAR on 7 out of 13 categories. Note that, overall on the entire test set (combining all question types) our model gives only slight improvement over GAR. The main reason for this is that the dataset is dominated by fill in the blank style questions and our model performs slightly poorly on such questions. However, since nearly 50% of the questions in the dataset are fill in the blank style questions even a small drop in the performance on these questions, offsets the gains that we get on other question types.

## 5.2 ENSEMBLE OF DIFFERENT MODELS

Since ElimiNet and GAR perform well on different question types we thought that taking an ensemble of these models should lead to an improvement in the overall performance. For a fair comparison, we also wanted to see the performance when we independently take an ensemble of $n$ GAR models and $n$ ElimiNet models. We refer to these as GAR-ensemble and ElimiNet-ensemble models. Each model in the ensemble is trained using a different hyperparameter setting and we use $n = 6$ (we did not see any benefit of using $n > 6$). We also report results of an ensemble of GAR-ensemble and ElimiNet-ensemble. The results of these experiments are summarized in Table 1. ElimiNet-ensemble performs better than GAR-ensemble and the final ensemble gives the best results. Overall, by taking an ensemble of the two models we get an accuracy of 47.2% which is 7% (relative) better than GAR and 2.8% (relative) better than GAR-ensemble.

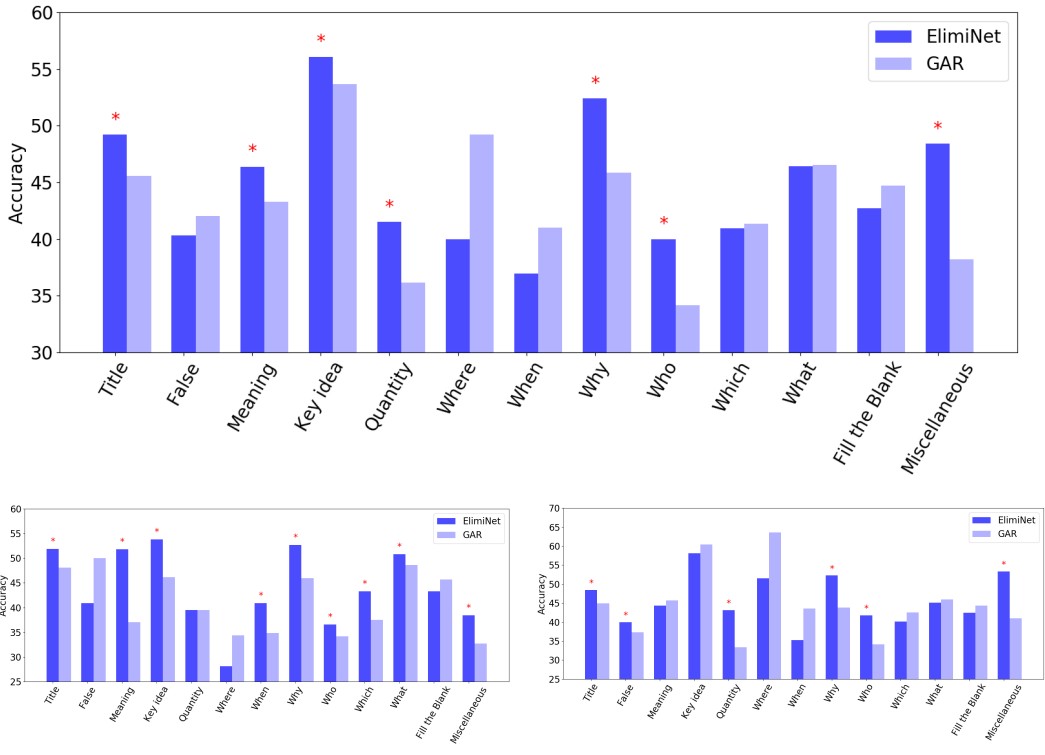

Figure 2: Performance of ElimiNet and Gated Attention Reader (GAR) on different question categories in RACE-Full (top), RACE-Mid (bottom left) and RACE-High (bottom right). The categories in which our model outperforms GAR are marked with *.

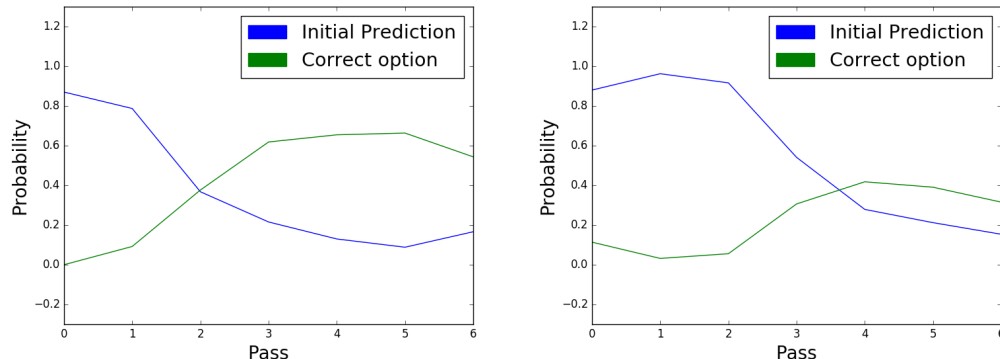

Figure 3: Change in the probability of correct option and incorrect option (initially predicted with highest score) over multiple passes of the *elimination* module. The two figures correspond to two different examples from the test set. The corresponding passage, question and options are given in the Appendix.

## 5.3 EFFECT OF SUBTRACT GATE

Next, we wanted to see if the subtract gate enables the model to learn better (by performing partial orthogonalization or alignment). To do this, we compare the accuracy with and without the subtract gate (in the latter case we just set the subtract gate to a vector of 1s). We observe that the accuracy of our model drops from 44.33% to 42.58% and we outperform the GAR model only in 3 out of 13 categories. This indicates that the flexibility offered by the subtract gate helps the model to learn better.

| Model | RACE |
|---|---|
| Stanford Attentive Reader | 43.3 |
| Gated Attention Reader (GAR) | 44.1 |
| **ElimiNet** | **44.5** |
| GAR Ensemble (6 models) | 45.9 |
| ElimiNet Ensemble (6 models) | 46.5 |
| GAR + ElimiNet | **47.2** |
| (ensemble of above 2 ensembles) | |

Table 1: Performance Comparison for Proposed Models

## 5.4 VISUALIZING PROBABILITY SCORES AFTER EACH ELIMINATION PASS

If the elimination module is indeed learning to eliminate options and align/orthogonalize the document representation w.r.t the uneliminated/eliminated options then we should see a shift in the probability scores as we do multiple passes of elimination. To visualize this, in Figure 3, we plot the probabilities of the correct option and the incorrect option which has the highest probability before passing through *elimination* module for two different test instances. The corresponding passage, question and options for these instances are provided in the Appendix. We observe that as we do multiple passes of elimination, the probability mass shifts from the incorrect option (blue curve) to the correct option (green curve). This in turn means that the elimination module is learning to align the document representation with the correct option (and hence its similarity increases) and moves it away from the incorrect option (hence its similarity decreases).

## 6 CONCLUSION

We focus on the task of Reading Comprehension with Multiple Choice Questions and propose a model which mimics how humans approach this task. Specifically, the model uses a combination of

elimination and selection to arrive at the correct option. This is achieved by introducing an elimination module which takes a soft decision as to whether an option should be eliminated or not. It then modifies the document representation to either align it with uneliminated options or orthogonalize it to eliminated options. The amount of orthogonalization or alignment is determined by two gating functions. This process is repeated multiple times to iteratively refine the document representation. We evaluate our model on the recently released RACE dataset and show that it outperforms current state of the art models on 7 out of 13 question types. Finally, using an ensemble of our elimination-selection approach with a state of the art selection approach, we get an improvement of 7% over the best reported performance on RACE dataset. As future work, instead of soft elimination we would like to use reinforcement learning techniques to learn a policy for hard elimination.

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

## A  PASSAGE EXAMPLES CORRESPONDING TO THE PLOTS IN FIGURE 3

**Passage:** If you're in charge of a project, the key to success is getting everyone to want to help you. As a director, I point, I suggest, I gently push the actors in the direction I want them to go. In the 1986 movie Nothing in Common, Jackie Gleason's character, Max Basner, gets fired (or unemployed) from his job as a clothing salesman. The scene, shot on a boat, shows Max's despair about being out of work. I was looking for some gesture that would allow Max to show his feelings. Jackie had far more experience at everything than I did, and at first I was frightened (very much afraid). What could I possibly tell "The Great One" about acting? Out of fear I decided to direct by suggestion, and I sat down with Gleason to talk about the scene. "So Max is sad, right?" I said. Gleason nodded. "And he's probably still carrying his pens with his name on them — the ones he used to hand out to his customers, right¿'Gleason nodded."So what would you want to do with the pens after you were fired?" He was silent for a moment. "Why don't I throw them overboard (or all over the boat)?" I stood up and turned toward the crew. " Hey, everybody, Jackie has a wonderful idea. Let's shoot it." After filming the scene, Gleason called me over and said with a smile, "Garry, what kind of wonderful idea am I going to have tomorrow?" You and your team can discover the answers to problems together. When there are no prizes or gold stars for who gets the solution (or the way to work out a problem) first, you'll all benefit (or get something of use or value) when everything turns out right.

**Question:** Why did Gleason call the director over and smile at him? That's because .....
**Option A**: Thought his wonderful idea was accepted by the latter.
**Option B**: Succeeded in hitting upon ( or think of ) a wonderful.
**Option C**: Was confident ( certain ) about his work the next day.
**Option D**: Appreciated the latter 's way of directing films.
**Correct Option :** D
**Initial Prediction:** B

**Passage:** What does it feel like to break a bone ? It's different for everyone, but the pain is often sharp . If the break is small, however, the person may not feel much pain at all. If you think that you or someone else has broken a bone, the most important things to do are to stay calm, make the hurt person comfortable, and call the doctor. Do not move the injured body part since movement could make it worse. To treat the break, the doctor will need to take an X-ray. This gives the doctor the information he or she needs to set the bone: to put it back to its normal place. If the bone is large or it is broken in more than one place, the doctor may need to use metal pins to set it. After the bone has been set, the next step is usually putting on a cast, the special, hard thing that will keep the bone in place for a month or two. Your bones are excellent at healing themselves. Broken bones will produce many new cells and tiny blood vessels. These cover both ends of the broken part, and close up the break until the bone is as whole and strong as before.

**Question:** Which of the following is the best title for the passage ?
**Option A**: "How to Know if a Bone is Broken"
**Option B**: "How Broken Bones Heal Themselves"
**Option C**: "Common Causes of Broken Bones"
**Option D**: "What You Should Know about Broken Bones".
**Correct Option :** D
**Initial Prediction:** B

Table 2: Examples corresponding to Figure 3 where *elimination* module decreases probability of wrong answer and increases probability of correct answer.

## B   QUESTION TYPES

| Categories | Example |
|---|---|
| **Why** | Question: Why didn't Alice expect there was also a gift for her?
A: The gifts from Santa Claus were usually for children.
B: The gift was forgotten many years ago.
C: The gift for her was bought by accident on the way.
D: The gifts for Katie were enough to share with her. |
| **Meaning in context of passage** | Question: What does the word " cup " in this passage refer to ?
A: Life
B: Happy things in life
C: Hot Chocolate
D: Tools to hold and contain life |
| **Title** | Question: Which of the following would probably be the best title for the passage ?
A: Webster 's dictionary
B: American Identity
C: The shaper of american english
D: Spelling differences between american and british english |
| **Key Idea** | Question: What is the california builder 's story intended to prove ?
A: The house structure is a key factor customers consider .
B: The more costly the house is , the better it sells .
C: An ocean view is much to the customers ' taste .
D: A good first impression increases sales . |
| **False** | Question: Which of the following is not true ?
A: English people drink a lot of tea .
B: Chinese food is different from british food
C: Both chinese people and english people like rice and fish.
D: English people like both chinese tea and indian tea. |
| **What** | Question: What color does sue like ?
A: White.
B: Red.
C: Yellow.
D: Green. |
| **Who** | Question: Who would pay for the cost of the reality TV shows according to the passage?
A: TV producers who make reality TV shows
B: TV actors who take part in reality TV shows
C: TV viewers who telephone reality TV shows
D: TV companies which broadcast reality TV shows |
| **Where** | Question: Where were the diamonds in the end?
A: They were in the bag.
B: They were on the plane.
C: They were in the bathroom.
D: They were taken away by the criminal. |
| **Which** | Question: Which of the following is true about the situation in iceland and thailand ?
A: Boys score higher in math tests than girls.
B: Girls are given more exceptional attention than boys .
C: More girls score higher in math than boys .
D: Boys are more creative than girls . |
| **Quantity** | Question: How many people with remarkable iq are mentioned in the passage ?
A: 3
B: 4
C: 5
D: 6 |
| **Fill in the blank** | Question: If you don't have a ........., you can never get anything.
A: Good face
B: Weak body
C: Strong body
D: Good father |

| | |
|---|---|
| **When** | Question: When did scientists fly there to watch the birth of the island?
A: Before the volcano broke out.
B: On 14th November 1963.
C: About four days after the volcano broke out.
D: In June 1967. |
| **Miscellanaous** | Question: How can we describe the writer's attitude toward life?
A: Practical
B: Passive
C: Disappointed
D: Positive |

Table 3: Question categories with examples

