# OpenReview forum: "ElimiNet: A Model for Eliminating Options for Reading Comprehension with Multiple Choice Questions"
_ICLR.cc/2018/Conference — Reject_

### Official Review · AnonReviewer3 · 2017-11-26
**Interesting problem but the results are not so good, hope the methods could be further improved in the future.**

**Rating:** 5
**Confidence:** 3

**Review:**

In this paper, a model is built for reading comprehension with multiple choices. The model consists of three modules: encoder, interaction module and elimination module. The major contributions are two folds: firstly, proposing the interesting option elimination problem for multi-step reading comprehension;  and secondly, proposing the elimination module where a eliminate gate is used to select different orthogonal factors from the document representations. Intuitively, one answer option can be viewed as eliminated if the document representation vector has its factor along the option vector ignored.

The elimination module is interesting, but the usefulness of “elimination” is not well justified for two reasons. First, the improvement of the proposed model over the previous state of the art is limited. Second, the model is built upon GAR until the elimination module, then according to Table 1 it seems to indicate that the elimination module does not help significantly (0.4% improvement).

In order to show the usefulness of the elimination module, the model should be exactly built on the GAR with an additional elimination module (i.e. after removing the elimination module, the performance should be similar to GAR but not something significantly worse with a 42.58% accuracy). Then we can explicitly compare the performance between GAR and the GAR w/ elimination module to tell how much the new module helps.

Other issues:

1) Is there any difference to directly use $x$ and $h^z$ instead of $x^e$ and $x^r$ to compute $\tilde{x}_i$? Even though the authors find the orthogonal vectors, they’re gated summed together very soon. It would be better to show how much “elimination” and “subtraction” effect the final performance, besides the effect of subtraction gate.

2) A figure showing the model architecture and the corresponding QA process will better help the readers understand the proposed model.

3) $c_i$ in page 5 is not defined. What’s the performance of only using $s_i$ for answer selection or replacing $x^L$ with $s_i$ in score function?

4) It would be better to have the experiments trained with different $n$ to show how multi-hop effects the final performance, besides the case study in Figure 3.

Minor issues:

1) In Eqn. (4), it would be better to use a vector as the input of softmax.

2) It would be easier for discussion if the authors could assign numbers to every equation.

---

### Official Review · AnonReviewer2 · 2017-11-27
**Competent elaboration of the Gated Attention Reader**

**Rating:** 5
**Confidence:** 3

**Review:**

This paper gives an elaboration on the Gated Attention Reader (GAR) adding gates based on answer elimination in multiple choice reading comprehension.  I found the formal presentation of the model reasonably clear the the empirical evaluation reasonably compelling.

In my opinion the main weakness of the paper is the focus on the RACE dataset.  This dataset has not attracted much attention and most work in reading comprehension has now moved to the SQUAD dataset for which there is an active leader board.  I realize that SQUAD is not explicitly multiple choice and that this is a challenge for an answer elimination architecture.  However, it seems that answer elimination might be applied to each choice of the initial position of a possible answer span.  In any case, competing with an active leader board would be much more compelling.

---

### Official Review · AnonReviewer1 · 2017-11-28

**Rating:** 4
**Confidence:** 4

**Review:**

This paper proposes a new reading comprehension model for multi-choice questions and the main motivation is that some options should be eliminated first to infer better passage/question representations.

It is a well-written paper, however, I am not very convinced by its motivation, the proposed model and the experimental results.

First of all, the improvement is rather limited. It is only 0.4 improvement overall on the RACE dataset; although it outperforms GAR on 7 out of 13 categories; but why is it worse on the other 6 categories? I don’t see any convincing explanations here.

Secondly, in terms of the development of reading comprehension models, I don’t see why we need to care about eliminating the irrelevant options. It is hard to generalize to any other RC/QA tasks. If the point is that the options can add useful information to induce better representations for passage/question, there should be some simple baselines in the middle that this paper should compare to. The two baselines SAR and GAR both only induce a representation from paragraph/question, and finally compare to the representation of each option. Maybe a simple baseline is to merge the question and all the options and see if a better document representation can be defined.

Some visualizations/motivational examples could be also useful to understand how some options are eliminated and how the document representation has been changed based on that.

---

### Decision · Program_Chairs · 2018-01-29
**ICLR 2018 Conference Acceptance Decision**

**Decision:**

Reject

**Comment:**

This paper provides a method for eliminating options in multiple-answer reading comprehension tasks, based on the contents of the text, in order to reduce the "answer space" a machine reading model must consider. While there's nothing wrong with this, conceptually, reviewers have questioned whether or not this is a particularly useful process to include in a machine reading pipeline, versus having agents that understand the text well enough to select the correct answer (which is, after all, the primary goal of machine reading). Some reviewers were uncomfortable with the choice of dataset, suggesting SQuAD might be a better alternative), and why I am not sure I agree with that recommendation, it would be good to see stronger positive results on more than one dataset. At the end of the day, it is the lack of convincing experimental results showing that this method yields substantial improvements over comparable baselines which does the most harm to this well written paper, and I must recommend rejection.